# ADAPTIVE INFERENCE-TIME COMPUTE:
# LLMS CAN PREDICT IF THEY CAN DO BETTER,
# EVEN MID-GENERATION

## ABSTRACT

Inference-time computation is a powerful paradigm to enhance the performance of large language models (LLMs), with Best-of-N sampling being a widely used technique. However, this method is computationally expensive, requiring both (1) an external reward model and (2) the generation of multiple samples. In this work, we introduce a new generative self-evaluation scheme designed to adaptively reduce the number of generated samples while maintaining or even improving performance. We use a generative reward model formulation, allowing the LLM to predict mid-generation the probability that restarting the generation will yield a better response. These predictions are obtained without an external reward model and can be used to decide whether or not to generate more samples, prune unpromising samples early on, or to pick the best sample. This capability is very inexpensive as it involves generating a single predefined token. Trained using a dataset constructed with real unfiltered LMSYS user prompts, Llama 3.1 8B's win rate against GPT-4 on AlpacaEval increases from 21% to 34% with 16 samples and math performance on GSM8K improves from 84% to 91%. By sampling only when the LLM determines that it is beneficial to do so and adaptively adjusting temperature annealing, we demonstrate that 74% of the improvement from using 16 samples can be achieved with only 1.2 samples on average. We further demonstrate that 50–75% of samples can be pruned early in generation with minimal degradation in performance. Overall, our methods enable more efficient and scalable compute utilization during inference for LLMs.

## 1 INTRODUCTION

As large language models (LLMs) continue to advance, delivering high-quality responses across diverse applications becomes increasingly important. One promising direction to enhance response quality is the strategic use of inference-time computation, particularly through methods like Best-of-N sampling Snell et al. (2024); Charniak & Johnson (2005); Cobbe et al. (2021), which selects the best response from multiple candidates. However, this method incurs substantial inference cost from querying an external reward model and producing a large, fixed number of samples.

In this work, we introduce a new reward modeling paradigm, which we denote as **capability-aware self-evaluations**. This paradigm allows for adaptive allocation of inference-time compute, aiming to reduce the computational overhead while maintaining or improving LLM performance across various domains. We demonstrate that LLMs can directly model the probability that restarting generation yields in a better response, enabling informed decisions about whether to continue generating a response, initiate new ones, as well as rank responses. These predictions are obtained by simply appending a predefined self-evaluation prompt to the partially or fully generated response and generating a single predefined token whose likelihood is used as the prediction. This is in contrast to preference-based reward models which can primarily only be used to rank responses.

Our self-evaluation method is highly cost-effective, requiring no external reward models and incurring only the minimal cost of generating a single token. In contrast, an external reward model inherently requires more memory and storage. Additionally, it is unable to reuse the KV cache obtained when generating the response and would have to process the input and response from scratch.

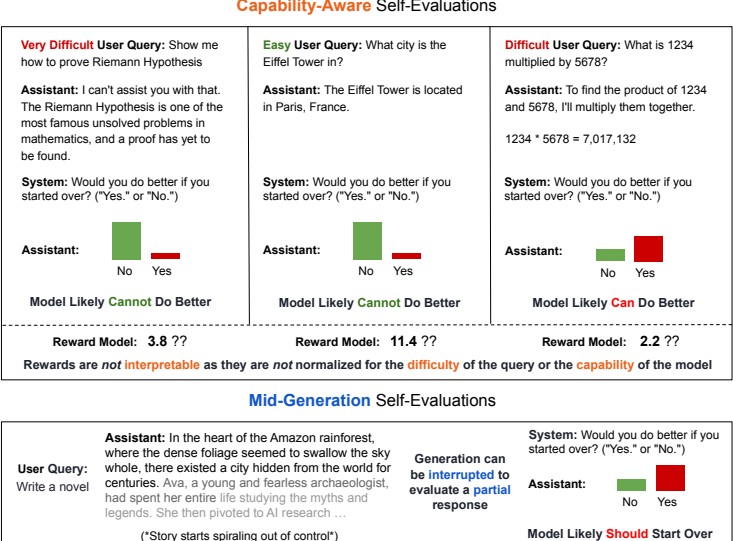

Figure 1: **Capability-Aware and Mid-Generation Self-Evaluations** enable adaptive inference-time compute strategies. They are obtained without an external reward model and can determine whether or not to generate more samples, prune unpromising samples early on, and pick the best sample.

To demonstrate adaptive inference-time compute allocation, we introduce two techniques: (1) adaptive sampling and (2) early pruning of unpromising samples. Adaptive sampling involves resampling a response for a given prompt until it is predicted that further samples will not yield additional improvements, thus conserving computation for complex tasks that will benefit from it. Furthermore, early pruning discards samples midway through generation if they are likely to result in suboptimal completions. These are not possible with standard reward models.

To give an LLM the ability to self-evaluate, one must construct an on-policy pairwise preference dataset with ties. In our experimental evaluation, we construct a dataset of approximately 30,000 preferences constructed using real unfiltered LMSYS (Chiang et al., 2024) user prompts and an existing reward model, ArmoRM (Wang et al., 2024) trained on roughly 1 million preferences or ratings. With this dataset, we fine-tune a Llama 3.1 8B Instruct (Dubey et al., 2024) model to self-evaluate and demonstrate significant performance improvements across both in-distribution and out-of-distribution tasks. Notably, as shown in Figure 4, the win rate against GPT-4 on AlpacaEval increases from 21% to 34% with 16 samples, and performance on held-out GSM8K math problems improves from 84% to 91%. We find that with adaptive sampling, using just 1.2 samples on average captures 74% of the improvement observed with 16 samples and 1.9 samples captures 84%. Additionally, early pruning can prevent 75% of unpromising samples from being fully generated, saving 56% of tokens generated with very minimal degradation in performance.

Our approach allows for more efficient and scalable use of compute resources during inference. By enabling models to dynamically allocate compute during inference based on task complexity and the model's capability, we optimize resource usage, ensuring efficiency in processing all types of prompts and tasks. This adaptability makes using inference-time compute far more practical and ready for the real world where LLMs are used for a wide variety of applications.

## 2 PRELIMINARIES AND NOTATION

An autoregressive language model generates a sequence $\mathbf{y} = (y_1, y_2, \ldots, y_T)$ given an input context $\mathbf{x}$ by predicting tokens sequentially. Assuming the model is parameterized by $\theta$, the conditional probability distribution of generating a sequence $\mathbf{y}$ given context $\mathbf{x}$ is

$$p_\theta(\mathbf{y}|\mathbf{x}) = \prod_{t=1}^{T} p_\theta(y_t|\mathbf{x}, y_{<t}), \tag{1}$$

with the convention $y_{<t} = (y_1, y_2, \ldots, y_{t-1})$. For ease of notation, we define $p_\theta(y_t|\mathbf{x}) := p_\theta(y_t|y_{<t}, \mathbf{x})$. For a vocabulary size $M$, the probability of predicting the $t$-th token $y_t$ is determined

using a softmax with temperature $\gamma$ on logit scores $z$ of all the tokens:

$$p_\theta(y_t|\mathbf{x}) = \frac{\exp(z_t/\gamma)}{\sum_{i=1}^M \exp(z_i/\gamma)}, \tag{2}$$

where $z_t = \text{logit}_\theta(y_t|\mathbf{x}, y_{<t})$. Higher values of $\gamma$ introduce more randomness; as the temperature $\gamma$ approaches zero, the distribution becomes concentrated on the token with the highest logit.

**Next-token prediction** is a typical approach used for pre-training and fine-tuning of LLMs. In particular, supervised fine-tuning (SFT) minimizes the cross-entropy loss between the model's predicted next token and the target token in a given sequence. Given a dataset $\mathcal{D} = \{(\mathbf{x}, \mathbf{y})\}$ of input context $\mathbf{x}$ and target response $\mathbf{y}$, the SFT loss is given by:

$$\mathcal{L}_{\text{SFT}}(\theta, \mathcal{D}) = -\mathbb{E}_{(\mathbf{x},\mathbf{y})\sim\mathcal{D}} \left[ \sum_{t=1}^{|\mathbf{y}|} \log p_\theta(y_t|\mathbf{x}, y_{<t}) \right]. \tag{3}$$

**On-policy pairwise preference dataset** is a preference dataset that consists of responses generated by a single model:

$$\mathcal{D}_{\text{preference}} = \{(\mathbf{x}, \mathbf{y}_1, \mathbf{y}_2, l)_i\}_{i=1}^N, \tag{4}$$

where $\mathbf{x}$ is the input, $\mathbf{y}_1$ and $\mathbf{y}_2$ are two responses generated by the model, and $l$ is the preference label that indicates the outcome of the comparison between $\mathbf{y}_1$ and $\mathbf{y}_2$:

$$l = \begin{cases} 1 & \text{if } \mathbf{y}_1 \text{ resulted in a } \textit{Win} \text{ and } \mathbf{y}_2 \text{ resulted in a } \textit{Loss } (\mathbf{y}_1 \succ \mathbf{y}_2), \\ 0 & \text{if } \mathbf{y}_1 \text{ and } \mathbf{y}_2 \text{ resulted in a } \textit{Tie } (\mathbf{y}_1 \approx \mathbf{y}_2), \\ -1 & \text{if } \mathbf{y}_1 \text{ resulted in a } \textit{Loss} \text{ and } \mathbf{y}_2 \text{ resulted in a } \textit{Win } (\mathbf{y}_1 \prec \mathbf{y}_2). \end{cases} \tag{5}$$

To simplify our analysis, we assume that the preference labels $l$ are based on an underlying reward function $r(\mathbf{x}, \mathbf{y})$ and a threshold $\epsilon$ that defines if one completion is more preferred:

$$l = \begin{cases} 1 & \text{if } r(\mathbf{x}, \mathbf{y}_1) - r(\mathbf{x}, \mathbf{y}_2) > \epsilon, \\ 0 & \text{if } |r(\mathbf{x}, \mathbf{y}_1) - r(\mathbf{x}, \mathbf{y}_2)| \le \epsilon, \\ -1 & \text{if } r(\mathbf{x}, \mathbf{y}_1) - r(\mathbf{x}, \mathbf{y}_2) < -\epsilon. \end{cases} \tag{6}$$

**Bradley-Terry (BT) (Bradley & Terry, 1952) reward models** consist of a base model $\theta_B$ and a shallow MLP reward head $\theta_R$. They predict a scalar reward score $\hat{R} = r_{\theta_B, \theta_R}(\mathbf{x}, \mathbf{y})$ given an input and response. They are initialized from a pretrained base model and a randomly initialized reward head, then trained on a dataset of $N$ examples, $\mathcal{D} = \{(\mathbf{x}, \mathbf{y}_{\text{win}}, \mathbf{y}_{\text{loss}})_i\}_{i=1}^N$, where $\mathbf{y}_{\text{win}}$ is the preferred response and $\mathbf{y}_{\text{loss}}$ is the dispreferred response. They are trained to predict a higher reward for $\mathbf{y}_{\text{win}}$ than for $\mathbf{y}_{\text{loss}}$ under the Bradley-Terry model. This is achieved by minimizing:

$$\mathcal{L}_{RM}(\theta_B, \theta_R, D) = -\mathbb{E}_{(\mathbf{x}, \mathbf{y}_{\text{loss}}, \mathbf{y}_{\text{win}})\sim D} \left[ \log\left(\sigma(r_{\theta_B, \theta_R}(\mathbf{x}, \mathbf{y}_{\text{win}}) - r_{\theta_B, \theta_R}(\mathbf{x}, \mathbf{y}_{\text{loss}}))\right) \right], \tag{7}$$

**Best-of-N** (Snell et al., 2024) is a widely used test-time compute approach to improve the performance of LLMs. Specifically, given a prompt, $N$ candidate responses from an LLM are scored using a reward model, and the highest-scoring response is filtered as the final response.

## 3 Adaptive Inference-Time Compute via Capability-Aware and Mid-Generation Self-Evaluations

A simple, yet effective method for allocation of test-time compute is Best-of-N, allowing the model to improve upon its responses over greedy sampling. However, the generation of a large, fixed number of samples is computationally expensive and the efficacy of this approach relies heavily on the quality of an external reward model, which can additionally incur substantial computational cost. In the following section, we address the cost and robustness of the reward model by establishing a general framework for self-evaluation via token prediction. We show how to train capability-aware and mid-generation self-evaluators to reduce the cost of generating a large, fixed number of samples.

## 3.1 Capability-Aware and Mid-Generation Self-Evaluations

In the classic reward modeling paradigm, a pre-trained LM is used as an initialization for reward modeling. However, this approach also relies on a newly added reward head to output the reward, which diverges from the token prediction task for which the LLM was trained, which can hurt performance for reward modeling. Additionally, generally preference-based reward models learn only to rank arbitrary responses. This may be limiting for comparing on-policy samples (such as those from Best-of-N) during inference to determine how good the response is with respect to the model's own capabilities. Furthermore, reward models are generally only able to evaluate full responses. Responses that are clearly unpromising very early on in generation still have to be generated until the end for evaluation, wasting computational resources. **Can we use an alternate paradigm to model rewards to address these limitations?**

We propose capability-aware and mid-generation self-evaluations, a paradigm in which we query the reward model to predict the probability that restarting generation will not yield a better response. This probability can be used to elicit adaptive test-time computation, where the model can dynamically allocate compute dependent on the difficulty of a query and its own capability. This is because to decide whether or not to allocate more compute to a given input, we need to know if it is fruitful to do so. For example, for easy queries, the model may have already outputted the best response it can, so resampling is unnecessary. Additionally, for difficult queries, sampling more may lead to a minor improvement in performance, leading to unfruitful resampling. We illustrate this paradigm in Figure 1. Modeling this probability effectively has three components: (1) reward modeling using token prediction to better leverage the pretrained model's existing knowledge and capabilities (2) relying on on-policy pairwise preferences and ties to model the probability that the model cannot generate a more preferred response (3) using truncated responses to model the probability that restarting generation will not yield a more preferred response. Traditional reward models can suffer at this task as they are model agnostic. Though these models can rank how different responses perform for a given task, they are unable to quantify how effective resampling would be (not capability aware). Furthermore, a desirable property for a reward model is to early stop the generation of samples that are not promising in order to give the model another chance without wasting compute.

**Reward Modeling with Token Prediction.** To obtain a self-evaluation of a response from an LLM, we simply append a predefined self-evaluation prompt $I$ to the generated response $\mathbf{y}$ and obtain a score in the form of the likelihood of a predefined token $t_{\text{good}}$ corresponding to the likelihood of the response being good. This approach allows us to acquire rewards without any external reward model, making it highly cost-effective as we can reuse the KV cache obtained during the generation of the response. Also, this leverages the existing zero-shot capability of the models to judge its own responses as a prior, which has been shown to be effective in works such as Madaan et al. (2023).

Formally, given a preference dataset $\mathcal{D}_{\text{pref}} = \{x, \mathbf{y}_{\text{good}}, \mathbf{y}_{\text{bad}}\}$, which contains input context $x$ and response pairs $y$, we can train a self-evaluation model by maximizing the likelihood of the good token $\log p_\theta(t_{\text{good}} \mid (\mathbf{x}, \mathbf{y}_{\text{good}}))$ for good responses $\mathbf{y}_{\text{good}}$, and the likelihood of the bad token $\log p_\theta(t_{\text{bad}} \mid (\mathbf{x}, \mathbf{y}_{\text{bad}}))$ for bad responses $\mathbf{y}_{\text{bad}}$. To achieve this, we minimize the SFT loss in eq. (3) on a modified variant of the preference dataset $\mathcal{D}_{\text{self-evaluation}}$

$$\mathcal{D}_{\text{self-evaluation}} = \{((\mathbf{x}, \mathbf{y}_{\text{good}}, I), t_{\text{good}})\} \cup \{((\mathbf{x}, \mathbf{y}_{\text{bad}}, I), t_{\text{bad}})\}. \tag{8}$$

During inference, we normalize the likelihood of $t_{\text{good}}$ as the score to rank responses:

$$\text{Score} = \frac{p_\theta(t_{\text{good}} \mid \mathbf{x}, \mathbf{y}, I)}{\sum_{t \in \{t_{\text{good}}, t_{\text{bad}}\}} p_\theta(t \mid \mathbf{x}, \mathbf{y}, I)}. \tag{9}$$

Given that the language model has a fixed vocab size $|V|$, probability mass can be spuriously assigned to tokens $t \notin \{t_{\text{good}}, t_{\text{bad}}\}$, resulting in this normalization to be desirable.

Note: it is beneficial to establish within the self-evaluation prompt that the LLM has to perform a classification task and that the target tokens are natural responses to effectively leverage the zero-shot capability of the model. We find that with a poorly designed self-evaluation prompt $I$, the model will overgeneralize during training and respond with $t_{\text{good}}$ or $t_{\text{bad}}$ for all queries, regardless of relevance. We need to keep the underlying policy or model unchanged when learning to self-evaluate. To do so, we need the model's prior distribution over tokens to be similar to what we expect after training where $p_\theta(t_{\text{good}} \cup t_{\text{bad}} \mid \mathbf{x}, \mathbf{y}, I) \approx 1$. Fortunately, this can easily be done by explicitly stating that the LLM should respond with only $t_{\text{good}}$ or $t_{\text{bad}}$ in the self-evaluation prompt $I$.

**The probability that the model cannot generate a more preferred response** can be easily derived from on-policy pairwise preferences and ties. It is the probability that the current sample $\mathbf{y}$ results in a *Win* or a *Tie* against another sample $\mathbf{y}'$. More formally:

$$P_{\mathbf{y}' \sim p_\theta(\mathbf{y}|\mathbf{x})} \left( r(\mathbf{x}, \mathbf{y}) - r(\mathbf{x}, \mathbf{y}') \geq -\epsilon \right) = P(Win \cup Tie \mid \mathbf{x}, \mathbf{y}). \tag{10}$$

Accounting for ties in reward modeling is especially important for on-policy pairwise data where responses coming from the same model are very likely to be similar. Ties commonly occur when generating responses from the same model (e.g. 40% of the time) and are even more common with simple tasks. Since ties indicate that the model cannot do better, they are crucial when learning to do capability-aware self-evaluations. If ties are not specified, it is also harder to model the reward as the model has to distinguish between extremely similar responses. Furthermore, since $(100 - 40)/2 = 30\%$ of samples result in a *Loss*, we see that the model can do significantly better on a query only roughly 30% of the time.

Notice that $P(Win \cup Tie \mid \mathbf{x}, \mathbf{y})$ monotonically increases with the sample's underlying reward $r(\mathbf{x}, \mathbf{y})$. This means that it can also be used to determine if one sample has a higher reward than another. Since inference-time compute strategies only care about being able to rank responses, it turns out that modeling $P(Win \cup Tie \mid \mathbf{x}, \mathbf{y})$ is more useful than $r(\mathbf{x}, \mathbf{y})$ since it can also be used inform decisions on whether or not to allocate more compute.

**The probability that restarting generation will not yield a more preferred response** can be used to evaluate the quality of a partial response $\mathbf{y}_{1:t}$. In the context of adaptive inference-time compute, if we find that this probability is low, we can stop spending additional inference resources on generating the remainder of the partial response, pruning this poor partial response. Again, in the context of on-policy pairwise preferences or ties, this probability is simply $P(Win \cup Tie)$ conditioned on a partial response $\mathbf{y}_{1:t}$:

$$P_{\substack{\mathbf{y}' \sim p_\theta(\mathbf{y}|\mathbf{x}) \\ \mathbf{y}_{t+1:T} \sim p_\theta(\mathbf{y}_{t+1:T}|\mathbf{x}, \mathbf{y}_{1:t})}} \left( r(\mathbf{x}, \mathbf{y}_{1:T}) - r(\mathbf{x}, \mathbf{y}') \geq -\epsilon \right) = P(Win \cup Tie \mid \mathbf{x}, \mathbf{y}_{1:t}). \tag{11}$$

**Modeling these probabilities with an on-policy pairwise preference dataset with ties.** To train LLMs to make capability-aware self-evaluations, we construct a dataset derived from an on-policy pairwise preference dataset with ties, where good responses are those that resulted in a *Win* or *Tie* ($\mathbf{y}_{\text{win}}$, $\mathbf{y}_{\text{tie}}$), and bad responses are those that resulted in a *Loss* ($\mathbf{y}_{\text{loss}}$). To train LLMs to make capability-aware self-evaluations mid-generation, we simply include the same examples but with responses randomly truncated ($\mathbf{y}_{\text{win,trunc}}$, $\mathbf{y}_{\text{tie,trunc}}$, $\mathbf{y}_{\text{loss,trunc}}$).

We use the following self-evaluation prompt and target tokens:

$$I = \text{'Would you do better if you started over? (``Yes.'' or ``No.'')'}$$

$$t_{\text{good}} = \text{'No'}, \quad t_{\text{bad}} = \text{'Yes'}$$

Our final dataset $\mathcal{D}_{\text{capability-aware}}$, used to minimize the SFT loss (eq. (3)), is constructed as:

$$\mathcal{D}_{\text{capability-aware}} = \{((\mathbf{x}, \mathbf{y}_{\text{win}}, I), t_{\text{good}})\} \cup \{((\mathbf{x}, \mathbf{y}_{\text{tie}}, I), t_{\text{good}})\} \cup \{((\mathbf{x}, \mathbf{y}_{\text{loss}}, I), t_{\text{bad}})\} \tag{12}$$

$$\cup \{((\mathbf{x}, \mathbf{y}_{\text{win,trunc}}, I), t_{\text{good}})\} \cup \{((\mathbf{x}, \mathbf{y}_{\text{tie,trunc}}, I), t_{\text{good}})\} \cup \{((\mathbf{x}, \mathbf{y}_{\text{loss,trunc}}, I), t_{\text{bad}})\}.$$

During inference, we compute the normalized likelihood of the $t_{\text{good}}$ ('No') token to score full or partial responses:

$$p_\theta(Win \cup Tie \mid \mathbf{x}, \mathbf{y}_{1:t}) = \frac{p_\theta(t_{\text{good}} \mid \mathbf{x}, \mathbf{y}_{1:t}, I)}{\sum_{t \in \{t_{\text{good}}, t_{\text{bad}}\}} p_\theta(t \mid \mathbf{x}, \mathbf{y}_{1:t}, I)} \tag{13}$$

## 3.2 Making Best-of-N Efficient and Scalable

Best-of-N is very computationally expensive, as it requires the generation of a large, fixed number of samples. This leads to wasted computation on queries that do not require more compute, and underutilization of compute on queries that could benefit from it. To remedy this and make Best-of-N more practical, we introduce two new primitives that leverage the capability-aware and mid-generation self-evaluations that we introduced in the last section. The first is adaptive sampling, where additional samples are allocated to a query only if they are predicted to be beneficial. The second is early pruning, where unpromising samples are stopped from being generated further to save inference computation.

---

**Algorithm 1** Adaptive Sampling and Annealing

---

**Require:** Input prompt $\mathbf{x}$, maximum samples $N_{\max}$, threshold $\tau$
1: Initialize $k \leftarrow 1$, $N_{\text{cum}} \leftarrow 0$, $\mathcal{S} \leftarrow \emptyset$
2: **while** $N_{\text{cum}} < N_{\max}$ **do**
3:     $N_k \leftarrow 1$ if $k = 1$ else $2^{k-2}$
4:     $N_{\text{cum}} \leftarrow N_{\text{cum}} + N_k$
5:     $\gamma_k \leftarrow 1 - 2^{-(k-1)}$
6:     Sample $N_k$ responses $\{\mathbf{y}_i\}_{i=1}^{N_k}$ using temperature $\gamma_k$
7:     **for** each $\mathbf{y}_i$ in $\{\mathbf{y}_i\}_{i=1}^{N_k}$ **do**
8:         Compute $p_i \leftarrow p_\theta(\textit{Win} \cup \textit{Tie} \mid \mathbf{x}, \mathbf{y}_i)$; add $(\mathbf{y}_i, p_i)$ to $\mathcal{S}$
9:     **end for**
10:    **if** Any $p_i > \tau$ **then break**
11:    **end if**
12:    $k \leftarrow k + 1$
13: **end while**
14: Select $\mathbf{y}^*$ from $\mathcal{S}$ with highest $p_i$
15: **Return** $\mathbf{y}^*$

---

### 3.2.1 ADAPTIVE SAMPLING AND ANNEALING

We introduce adaptive sampling as a technique to allocate inference-time compute only when it is beneficial to do so. We further introduce exponentially increasing parallel sampling to mitigate the main disadvantage of adaptive sampling, latency. Finally, we introduce a temperature annealing strategy to balance exploration and exploitation while adaptively sampling and boost efficiency.

**Resampling until meeting a threshold.** We adaptively sample by resampling only when the model predicts it can produce a better response. We do so by computing the likelihood that the model cannot generate a better response $p_\theta(\textit{Win} \cup \textit{Tie} \mid \mathbf{x}, \mathbf{y})$ for every sample. If this probability exceeds a predefined threshold $\tau$ for any sample, we stop resampling and select the sample with the highest $p_\theta(\textit{Win} \cup \textit{Tie} \mid \mathbf{x}, \mathbf{y})$ as the final response. Otherwise, we resample, repeating this process until the threshold is met or a maximum number of samples $N_{\max}$ is reached. This approach concentrates computational resources on prompts where improvement is likely, avoiding unnecessary sampling when further gains are improbable.

**Increasing number of parallel samples exponentially.** To reduce latency, we sample in exponentially increasing batch sizes. Specifically, for the $k$-th sampling iteration, the batch size $N_k$ is defined as:

$$N_1 = 1, \quad N_k = 2^{k-2} \quad \text{for } k > 1. \tag{14}$$

This ensures that the cumulative number of samples by the $k$-th iteration is $2^{k-1}$. This exponential increase minimizes latency, reducing the number of iterations needed to meet $N_{\max}$, while allowing for enough self-evaluations to determine if larger batches of samples are necessary.

**Temperature annealing schedule based on number of samples generated so far.** To balance exploitation and exploration, we vary the temperature $\gamma$. The temperature for the $k$-th iteration is given by:

$$\gamma_k = 1 - 2^{-(k-1)}. \tag{15}$$

Initially, the temperature starts low (e.g., $\gamma_1 = 0$) to prioritize high-probability responses. As $k$ increases, $\gamma_k$ quickly approaches 1, encouraging more diverse and creative sampling. This annealing schedule allows the model to first focus on exploiting the most likely responses, then explore alternative options as more samples are generated.

### 3.2.2 EARLY PRUNING OF UNPROMISING SAMPLES

One downside to adaptive sampling requires the generation of samples in series, which introduces latency. However, can we still leverage parallel sampling to avoid increasing the latency of response generation but still enable adaptive test-time compute allocation? One approach to reducing computational costs in parallel generation is to early prune unpromising samples based on mid-generation self-evaluations.

**When to prune.** Pruning too early risks discarding samples that could improve, while pruning too late offers minimal savings. Thus, we balance this trade-off when selecting the fixed number of initial tokens (e.g., 64 or 128) before making any pruning decisions.

**Which samples to prune.** After generating an initial number of tokens, we compute the intermediate self-evaluations for each partially generated sample. After ranking samples by the resulting scores, we stop the generation of the bottom $x\%$ (e.g., 50% or 75%) to conserve computation. This ensures that only the most promising partial samples continue to be generated.

## 4 EXPERIMENTS

We evaluate our methods by first examining the sample efficiency of self-evaluation. We then evaluate its impact on optimizing inference-time compute with adaptive sampling and early pruning of unpromising samples.

### 4.1 TRAINING DATA AND EVALUATIONS

**Construction of On-Policy Pairwise Preference Dataset with Ties.** Training reward models typically requires human-labeled preferences, which are costly to obtain, especially for on-policy data generated by the model being trained. To mitigate this, we utilize an existing reward model, ArmoRM (Wang et al., 2024) trained on approximately 1,000,000 preferences. This reward model serves as our underlying reward $r(\mathbf{x}, \mathbf{y})$.

We conduct our experiments using the Llama 3.1 8B Instruct model, which is fine-tuned on a preference dataset of 50,000 real user prompts from LMSYS (Chiang et al., 2024) and pairs of on-policy responses (sampled from the same Llama 3.1 8B Instruct model). These responses are scored using ArmoRM, the underlying reward. Preferences or ties between responses are determined using a threshold $\epsilon$ of 0.01 on the reward difference, resulting in an on-policy pairwise preference dataset with approximately 40% ties.

**Evaluation Protocol** We evaluate the performance of our self-evaluation model and adaptive test-time compute approaches in two domains: (1) the AlpacaEval (Dubois et al., 2024) benchmark and (2) the GSM8K dataset (Cobbe et al., 2021). **AlpacaEval 2.0** is an automatic benchmark that compares model responses to those generated by GPT-4 across approximately 800 representative prompts. The final metric is the win rate against GPT-4, adjusted to increase correlation with human preferences (Spearman correlation of 0.98). While highly correlated with human judgments, this metric is relative—it measures success based on outperforming another model rather than achieving absolute human satisfaction. **GSM8K** is a collection of 8.5K grade school math word problems involving multi-step reasoning and basic arithmetic operations. We use GSM8K as it is a popular benchmark for reasoning as well as absolute measure of performance that is not relative to another LLM. This is particularly useful in evaluating adaptive sampling where it allocates compute resources to queries that benefit from it.

### 4.2 PERFORMANCE AND SAMPLE EFFICIENCY

| Benchmark | Random (1 sample) | Zero-Shot (LLM-as-a-Judge) | Bradley-Terry Reward Model | Capability-Aware $p_\theta(\textit{Win} \cup \textit{Tie})$ | Underlying $r(\mathbf{x}, \mathbf{y})$ |
|---|---|---|---|---|---|
| AlpacaEval | 21.2 | 24.4 | 33.2 | **33.8** | 36.3 |
| GSM8K | 84.2 | 86.7 | 87.7 | **91.0** | 92.6 |

Table 1: **Best-of-16 Performance on AlpacaEval (Win Rate vs GPT-4, %) and GSM8K (%)** using capability-aware self-evaluations using token prediction, LLM-as-a-Judge which also uses token prediction but without finetuning, the commonly used Bradley-Terry external reward model, and the underlying reward model used to create our experimental preference data.

**Baselines.** To evaluate performance with Best-of-N, we use two baselines. The Zero-Shot (LLM-as-a-Judge) baseline uses Llama 3.1 8B Instruct without additional training. We prompt and get scores for zero-shot predictions in the exact same manner that we do for capability-aware self-evaluations. The Bradley-Terry Reward Model also uses Llama 3.1 8B Instruct as the base model and is trained using the same on-policy pairwise preference dataset with ties.

As shown in Table 1, the zero-shot approach performs modestly on both benchmarks, confirming that LLMs have a non-trivial ability to self-evaluate. The Bradley-Terry reward model unsurpris-

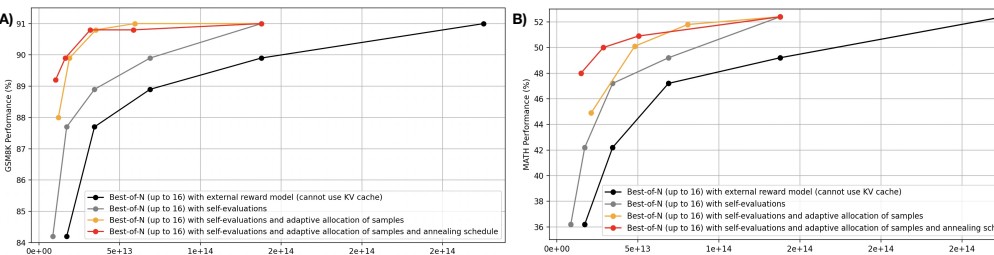

Figure 2: **FLOPs vs Performance for Adaptive Sampling and Annealing** In two domains (A: GSM8K, and B: MATH), we find savings in FLOPs from (1) self-evaluation, (2) adaptive allocation of samples and (3) a temperature annealing schedule.

ingly does much better. Our token prediction method outperforms both. On AlpacaEval, it achieves a 33.8% win rate against GPT-4 using Best-of-16 sampling, slightly surpassing the Bradley-Terry model's 33.2% and nearing the underlying reward model's 36.3% (trained on 1 million preferences). On GSM8K, token prediction attains 91.0% accuracy, significantly outperforming the Bradley-Terry model's 87.7%. This suggests that our method is more sample-efficient and generalizes better, especially on queries that might be uncommon among real user prompts like those in GSM8K.

These results demonstrate that token prediction effectively leverages the pre-trained model's priors for self-evaluation, approaching the performance of the larger Llama 3.1 70B, which achieves 38.1% on AlpacaEval and 95.1% on GSM8K.

We also examined other probabilities to model with token prediction, specifically $p_\theta(Win)$ or $p_\theta(Win \mid \neg Tie)$. Our findings, as shown in Table 2, indicate that including or removing ties does not significantly impact performance, but modeling $p_\theta(Win)$ is significantly more difficult as it forces the model to distinguish between samples that result in wins or ties, which are both cases in which the model performs relatively well for its capability.

Notably, using Best-of-N sampling with our methods allows the Llama 3.1 8B model to approach the performance of Llama 3.1 70B, a model roughly $10\times$ larger that gets 38.1% and 95.1% respectively. This highlights the efficacy of Best-of-N sampling in enhancing model performance. In the following section, we address the expensive nature of this method with adaptive compute strategies.

### 4.3 EFFICIENCY AND SCALING OF ADAPTIVE SAMPLING AND ANNEALING

**Metrics for Evaluation** Following prior work (Hoffmann et al., 2022; Pope et al., 2022; Chen et al., 2023), we study the efficiency as a measure of floating-point operations per second (FLOPS), which is proportional to the number of inference tokens generated. We additionally consider the inference latency through the number of sequential calls/batches and the wall time.

We evaluate the efficiency gains and performance trade-offs of our adaptive sampling and annealing strategy on the GSM8K and MATH datasets. Our objective is to retain the performance benefits of Best-of-16 sampling while significantly reducing the average number of FLOPs utilized. In this setting, we want to evaluate the methods' ability to allocate samples when necessary. Therefore, our experiments always use capability-aware self-evaluations to select the best sample so that the final selection method is not a confounding factor.

As a baseline, we first assess the performance of Best-of-N. The best sample is selected using self-evaluation with token prediction. Figure 2 shows that increasing the number of samples from 1 to 16 incrementally improves the GSM8K Pass@1 accuracy from 84.2% to 91.0%. This represents the maximum achievable performance with our method, which we define as 100% of the maximum improvement. However, generating a fixed large number of samples per query is expensive.

To mitigate this, we introduce adaptive sampling controlled by a threshold $\tau$ and a maximum number of samples $N_{\max}$. Initially, we test adaptive sampling using the underlying reward model to decide whether to generate additional samples. As shown in Figure 2, this approach does not significantly outperform random selection. While the underlying reward model effectively ranks responses, it does not consider the model's capability to improve upon its own outputs, substantially limiting its effectiveness in informing resampling decisions.

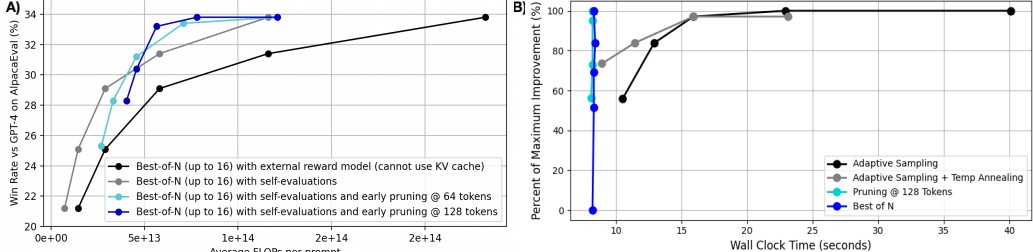

Figure 3: **FLOPs and Latency vs Performance for Pruning** In Alpaca Eval, we find (A) savings in FLOPs from pruning with (B) no increase in latency (in Wall Time).

In contrast, our adaptive sampling method utilizing capability-aware self-evaluations via token prediction yields substantial efficiency gains. Table 4 demonstrates that by adjusting the threshold $\tau$ for the win-or-tie probability $p_\theta(\textit{Win} \cup \textit{Tie} \mid \mathbf{x}, \mathbf{y})$, we can control the average number of samples. For instance, setting $\tau = 0.98$ results in 97.1% of the maximum performance while using only an average of 4.1 samples, compared to the 16 samples required for maximum performance.

Moreover, incorporating the annealing schedule further enhances performance, particularly at lower thresholds. As illustrated in Table 3, with annealing, we achieve 73.5% of the maximum improvement using an average of just 1.2 samples when $\tau = 0.92$. This indicates that our annealing strategy effectively balances exploitation and exploration during sampling.

Our adaptive sampling approach offers significant inference computational savings compared to Best-of-N. By adjusting $\tau$ and $N_{\max}$, we can tune the trade-off between performance and efficiency. The latency introduced by adaptive sampling is minimal due to the exponentially increasing batch sizes as seen in Figure 3, and a higher performance can be tradeoff with some additional latency. However, can we construct an approach that has no additional latency altogether?

## 4.4 EFFICIENCY GAINS OF PRUNING UNPROMISING SAMPLES

We evaluate the efficiency gains from early pruning of unpromising samples on the AlpacaEval benchmark. We begin with 16 samples and prune from them. Our primary metrics are the percent of maximum improvement achieved and the average FLOPs per prompt. We additionally study the Wall Clock Time (seconds) to estimate the latency of sampling approaches.

Our experiments control two variables: the number of tokens generated before making pruning decisions (64 or 128 tokens), and the percentage of samples pruned (e.g., pruning 75%). Again, the capability-aware self-evaluations are used the final sample selection method.

**Impact of Evaluation Timing on Performance.** As shown in Figure 3, pruning at 64 tokens improves performance over random pruning at 0 tokens. For instance, pruning 50% of samples at 64 tokens achieves a win rate of 33.4%, closely approaching the maximum win rate of 33.8% without pruning. Pruning 75% of samples at 128 tokens yields even better results, attainng a win rate of 33.2%, nearly matching the maximum performance without pruning. This improvement is expected, as longer partial responses provide more context for accurate self-evaluation.

Overall, there are substantial savings if one prunes at the right time and the right number of samples. In terms of computational cost measured by FLOPs, pruning 8 of the 16 samples at 128 tokens significantly reduces the average FLOPs per prompt by 2x as shown in Figure 5, resulting in significant efficiency gains, without any additional inference latency.

## 5 RELATED WORK

**Reward Modeling in RLHF.** Traditionally, reward models (RMs) within Reinforcement Learning with Human Feedback (RLHF) have been trained using discriminative techniques, often drawing on ranking models like the Bradley-Terry (BT) framework (Bradley & Terry, 1952). These RMs are generally treated as binary classifiers, distinguishing between preferred and dispreferred completions for a given prompt (Stiennon et al., 2020; Ouyang et al., 2022). The preference signal is predicted by generating a continuous score, typically derived from a linear head appended to an autoregressive language model. An alternative line of work departs from this discriminative paradigm

by directly modeling preferences through joint probability estimations, as seen in methods that predict the likelihood of one response being preferred over another (An et al., 2023).

Recent advancements in RLHF have introduced implicit reward models that circumvent the necessity for a distinct reward classifier by learning the distribution of preferred and dispreferred completions via objectives like Direct Preference Optimization (DPO) (Rafailov et al., 2023) and Implicit Preference Optimization (IPO) (Gheshlaghi Azar et al., 2023). These methods embed preference learning directly into the policy optimization process, unlike explicit preference modeling.

"LLM as a Judge" (Zheng et al., 2023) is another form of reward modeling where the model is prompted to act as evaluators without additional finetuning. Despite the potential of these methods, even advanced models like GPT-4o underperform when compared to dedicated RMs in more complex evaluations, such as those found in the RewardBench benchmark (Lambert et al., 2024).

**Techniques for Inference-time Compute.** During inference, the integration of a reward model with a proposal distribution (LLM) can be employed to refine the output responses to a given prompt. One notable paradigm in this context is Self-Consistency Wang et al. (2023), which is designed for factual queries with extractable answers. In this approach, the language model selects the response it generates with the highest frequency across multiple samples. Optimizations such as Early Stopping Self-Consistency have been proposed, which terminate the sampling process early if a subset of responses shows strong consistency. However, these approaches face limitations due to their dependence on identifying the most "consistent" response from a large pool of discrete answers, restricting their applicability to tasks like multiple-choice or mathematical problem-solving.

In addition, search algorithms such as Best-of-N and Beam Search have been explored in works such as Snell et al. (2024), which leverage reward models to select the most promising candidate samples in reasoning tasks. In this work, we examine how to enhance the efficiency of token generation relative to the accuracy of the outputs, with the complexity of the query dictating the amount of inference compute allocation. Furthermore, for multi-step reasoning tasks, this work presupposes that the problem can be decomposed into discrete semantic steps—a potentially strong assumption in domains outside of well-structured fields like mathematics.

**Generative Verifiers.** Concurrent work, such as Zhang et al. (2024); Ankner et al. (2024), demonstrate that token-based reward modeling outperforms traditional reward modeling techniques. We further introduce capability-aware self-evaluation, which allows a model to understand its own generation capability to enable dynamic allocation of computational resources during inference. We also introduce mid-generation self-evaluations, allowing for the pruning of unpromising samples.

## 6 DISCUSSION, CONCLUSION, AND LIMITATIONS

As large language models (LLMs) evolve, enhancing response quality through inference-time computation becomes critical. Best-of-N sampling, a traditional approach, generates multiple response candidates and selects the best, but it incurs high computational costs due to its reliance on external reward models and fixed sample sizes. This work introduces a cost-effective alternative: capability-aware self-evaluations performed mid-generation. By appending a self-evaluation prompt to responses, LLMs predict the likelihood of generating a better response without requiring external reward models. Two adaptive inference-time techniques—adaptive sampling and early pruning—are also proposed, allowing LLMs to resample or discard suboptimal responses during generation dynamically. These methods significantly improve efficiency, with fewer samples yielding performance gains similar to a larger set, and early pruning reducing unnecessary computation. Experimental results demonstrate notable improvements in performance across tasks, achieving a 34% win rate against GPT-4 on AlpacaEval with 16 samples and increasing accuracy on GSM8K math problems from 84% to 91%. Overall, this approach optimizes inference-time compute allocation, making it more scalable and practical for diverse real-world applications.

The main limitation of our method is that adaptive sampling introduces latency, which otherwise would not be a problem with only parallel sampling. We minimized this with exponentially increasing batch sizes. We also introduced early pruning to save computation even in parallel sampling. For future work, it may be possible to predetermine the number of samples one needs to allocate to a given query based on the probabiliy that samples result in a *Tie*. While this would be significantly less efficient than adaptive sampling, it would eliminate latency for adaptive sampling. It may be possible to do different types of search with active capability-aware self-evaluations. Beam search or other similar types of search may be possible on general prompts with this new capability.

## 7 Reproducability Statement

For reproducability, we provide the following details so that readers can replicate our results. Firstly, we provide details on how the dataset is constructed in sections 3 and 4 to enable Capability-Aware and Mid-Generation Self-Evaluations for any model. Additionally, we provide algorithm pseudocode as seen in algorithm 1, giving the reader transparency in how to replicate the adaptive sampling and annealing algorithm. Additionally, we provide details on how the dataset is constructed in sections 3 and 4 to enable Capability-Aware and Mid-Generation Self-Evaluations for any model. Finally, we provide evaluation details in section 4. For the camera ready, we hope to open-source the model we train and release a public Github implementation.

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

# A  APPENDIX

## A.1  EFFICIENCY WITH THE NUMBER OF SAMPLES GENERATED

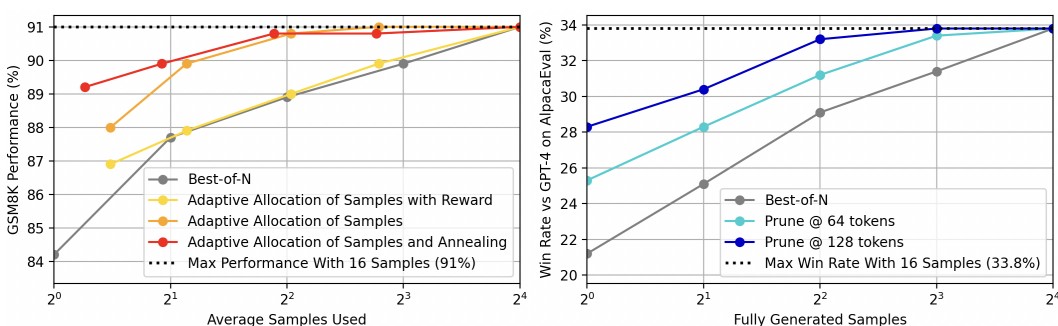

Figure 4: **Adaptive Inference-Time Compute**: Strategies such as adaptive sampling (left) and early pruning (right) make compute utilization during inference far more efficient and scalable.

## A.2  COMPARING LIKELIHOODS

We briefly compare alternative probabilities one might consider to model using an on-policy pairwise preference dataset with ties.

| Benchmark (Best-of-16) | $p_\theta(Win)$ | $p_\theta(Win \mid \neg Tie)$ | $p_\theta(Win \cup Tie)$ |
|---|---|---|---|
| Win Rate vs GPT-4 on AlpacaEval (%) | 27.7 | **30.1** | 29.6 |
| GSM8K Pass @ 1 (%) | 86.9 | 88.5 | **88.6** |

Table 2: Performance of different likelihoods with token prediction trained on a much smaller dataset of roughly 600 preferences.

## A.3  ADAPTIVE SAMPLING RESULTS

We show adaptive sampling performance with various aspects of the algorithm removed.

| Win-or-Tie Probability Threshold | 0.92 | 0.96 | 0.98 | 0.99 | 1.00 |
|---|---|---|---|---|---|
| Average Samples Used | **1.2** | **1.9** | **3.7** | **6.8** | **16.0** |
| Average Batches Used (Latency) | 1.1 | 1.4 | 2.0 | 2.9 | 5.0 |
| Average Wall Clock Time in seconds (Latency) | 8.92 | 11.4 | 15.9 | 23.1 | 39.5 |
| GSM8K Pass@1 (%) | 89.2 | 89.9 | 90.8 | 90.8 | 91.0 |
| Percent of Maximum Improvement | **73.5** | **83.8** | **97.1** | **97.1** | **100.0** |

Table 3: **Adaptive Sampling and Annealing** performance gains and efficiency on GSM8K. We find that latency (in the number of batches used and Wall Time) is comparable to Best-of-16 for smaller threshold values.

| Win-or-Tie Probability Threshold | 0.92 | 0.96 | 0.98 | 0.99 | 1.00 |
|---|---|---|---|---|---|
| Average Samples Used | **1.4** | **2.2** | **4.1** | **6.9** | **16.0** |
| Average Batches Used (Latency) | 1.3 | 1.6 | 2.0 | 2.9 | 5.0 |
| Average Wall Clock Time in seconds (Latency) | 10.5 | 12.9 | 15.9 | 22.9 | 40.1 |
| GSM8K Pass@1 (%) | 88.0 | 89.9 | 90.8 | 91.0 | 91.0 |
| Percent of Maximum Improvement | **55.9** | **83.8** | **97.1** | **100.0** | **100.0** |

Table 4: Performance on GSM8K with adaptive sampling using capability-aware self-evaluations. This is without the annealing schedule.

| Reward Threshold | 0.119 | 0.133 | 0.150 | 0.163 | Inf |
|---|---|---|---|---|---|
| Average Samples Used | **1.4** | **2.2** | **4.1** | **6.9** | **16.0** |
| Average Batches Used (Latency) | 1.2 | 1.4 | 2.0 | 2.8 | 5.0 |
| Average Wall Clock Time in seconds (Latency) | 9.4 | 11.3 | 15.9 | 22.4 | 39.2 |
| GSM8K Pass@1 (%) | 86.9 | 87.9 | 89.0 | 89.9 | 91.0 |
| Percent of Maximum Improvement | **39.7** | **54.4** | **70.6** | **83.8** | **100.0** |

Table 5: Performance on GSM8K with adaptive sampling using the underlying reward model.

| Samples Used | **1** | **2** | **4** | **8** | **16** |
|---|---|---|---|---|---|
| GSM8K Pass@1 (%) | 84.2 | 87.7 | 88.9 | 89.9 | 91.0 |
| Average Batches Used (Latency) | **1.0** | **1.0** | **1.0** | **1.0** | **1.0** |
| Average Wall Clock Time in seconds (Latency) | **8.2** | **8.3** | **8.3** | **8.4** | **8.3** |
| Percent of Maximum Improvement | **0.0** | **51.5** | **69.1** | **83.8** | **100.0** |

Table 6: Performance on GSM8K with Best-of-N at varying number of samples.

## A.4 EARLY PRUNING RESULTS

We show the results associated with early pruning shown in Figure 4 as well as tokens generated.

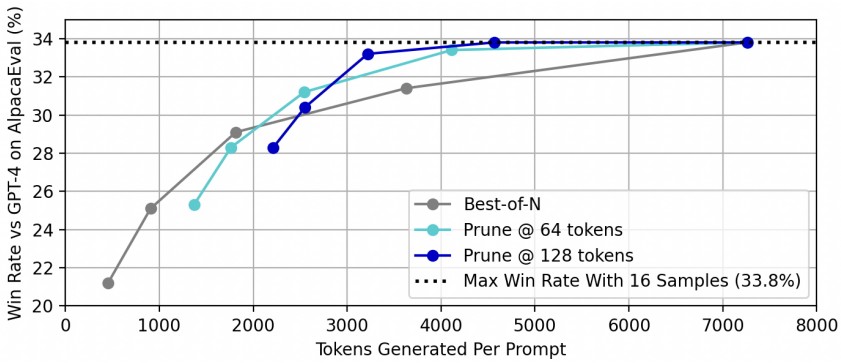

Figure 5: **Early Pruning** performance gains and efficiency on AlpacaEval.

| Win Rate vs GPT-4 on AlpacaEval (%) | Samples Used (Pruned Samples) | | | | |
|---|---|---|---|---|---|
| | 1 (Prune 15) | 2 (Prune 14) | 4 (Prune 12) | 8 (Prune 8) | 16 (Prune 0) |
| Prune @ 0 tokens (Random) | 21.2 | 25.1 | 29.1 | 31.4 | 33.8 |
| Prune @ 64 tokens | 25.3 | 28.3 | 31.2 | 33.4 | 33.8 |
| Prune @ 128 tokens | 28.3 | 30.4 | 33.2 | 33.8 | 33.8 |
| No Pruning | 33.8 | 33.8 | 33.8 | 33.8 | 33.8 |

Table 7: Win Rate vs GPT-4 on AlpacaEval with varying pruning strategies.

| Percent of Maximum Improvement (%) | Samples Used (Pruned Samples) | | | | |
|---|---|---|---|---|---|
| | 1 (Prune 15) | 2 (Prune 14) | 4 (Prune 12) | 8 (Prune 8) | 16 (Prune 0) |
| Prune @ 0 tokens (Random) | 0.0 | 31.0 | 62.7 | 81.0 | 100.0 |
| Average Batches Used (Latency) | **1.0** | **1.0** | **1.0** | **1.0** | **1.0** |
| Average Wall Clock Time in seconds (Latency) | **7.9** | **8.1** | **8.1** | **8.0** | **8.2** |
| Prune @ 64 tokens | 32.5 | 56.3 | 79.4 | 96.8 | 100.0 |
| Prune @ 128 tokens | 56.3 | 73.0 | 95.2 | 100.0 | 100.0 |
| Average Batches Used (Latency) | **1.0** | **1.0** | **1.0** | **1.0** | **1.0** |
| Average Wall Clock Time in seconds (Latency) | **8.1** | **8.2** | **8.2** | **7.9** | **8.3** |
| No Pruning | 100.0 | 100.0 | 100.0 | 100.0 | 100.0 |

Table 8: Percent of Maximum Improvement achieved on AlpacaEval with different pruning strategies.

| Tokens Generated per Prompt | Samples Used (Pruned Samples) | | | | |
|---|---|---|---|---|---|
| | 1 (Prune 15) | 2 (Prune 14) | 4 (Prune 12) | 8 (Prune 8) | 16 (Prune 0) |
| Prune @ 0 tokens (Random) | 454 | 907 | 1,815 | 3,629 | 7,259 |
| Prune @ 64 tokens | 1,370 | 1,763 | 2,544 | 4,113 | 7,259 |
| Prune @ 128 tokens | 2,214 | 2,551 | 3,220 | 4,566 | 7,259 |
| No Pruning | 7,259 | 7,259 | 7,259 | 7,259 | 7,259 |

Table 9: Average tokens generated per prompt on AlpacaEval with different pruning strategies.

