# OpenReview forum: "Adaptive Inference-Time Compute: LLMs Can Predict if They Can Do Better, Even Mid-Generation"
_ICLR.cc/2025/Conference — Submitted to ICLR 2025_

### Official Review · Reviewer_Pqum · 2024-10-31

**Soundness:** 3
**Presentation:** 3
**Contribution:** 2
**Rating:** 5
**Confidence:** 3

**Summary:**

The paper proposes a new strategy to improve the inference of LLMs, involving two key components: adaptive sampling (selecting the N in best-of-N as a function of the prompt) and early pruning (pruning unpromising mid-generations). This relies on a new ability from LLMs; given a prompt and a potentially unfinished generation, they can learn to detect whether they will be better with an additional sampling round. This ability is trained explicitly in supervised way, with feedbacks obtained from an external RM; the key point being that this RM is then not required during deployment. This strategy improves performances on math and gsm8k benchmarks.

**Strengths:**

* The paper is clearly written, properly highlighting the different ideas and results.
* Improving inference strategies for LLms is an important topic for the ICLR community, with potential significant impact.
* The two contributions, the discovered "trainable self-evaluations" and its applications (adaptive sampling and early pruning) make sens.
* The experiments, althought succinct (see weakness), show convincing performances on standard benchmarks.

**Weaknesses:**

* The main limitation of the work is that the proposed ideas are arguably small improvements over existing baselines. For example, the paper does not mention "process-based reward models" or the "reward-guided decoding" literature (such as MCTS), that seeks the same objective. The only benefit of the paper would be to include those abilities inside the LLM itself, thus removing the need for an external value network.

* In terms of practicality, the need to generate sequentially is a key limitation in real-time deployment, thus hindering some of the efficiency benefits. Though, I acknowledge that the idea of "exponentially increased batch size" is an interesting but limited answer.

* With your training, and given a prompt and generation, you actually train the model to estimate the quantile of this generation according to the external reward model; a discussion on this topic would be needed. Then similarly, it would be interesting to consider extension where, be given N>2 generations, your model learns to detect whether the generation is actually the best-of-N.

* Some ablations are missing. For example, could you provide the performances of applying temperatre annealing to basic Best-of-N? What if you train an external RM to evaluate the quantile on mid-generations? Or about a simple strategy that would detect, be given the prompt, its complexity, and then allocate automatically a prompt-aware compute.

* I fully agree that "We need to keep the underlying policy or model unchanged when learning to self-evaluate." (l.232) Though I disagree with the next sentence: "To do so, we need the model’s prior distribution over tokens to be similar to what we expect after training" (l.233). Could you clarify this?
From my understanding, finetuning for self evalaution should decrease other abilities because of catastrophic forgetting.

## Nit

* The fig1 actually shows the opposite of what the legend "Rewards are not interpretable" states; indeed, actually the reward provides the lowest score to the most unconfident generation.

* Related work: some papers that may be worth discussing
"Don't throw away your value model! Generating more preferable text with Value-Guided Monte-Carlo Tree Search decoding"
"ARGS: Alignment as Reward-Guided Search"
"Efficient Controlled Language Generation with Low-Rank Autoregressive Reward Models"
"Critic-Guided Decoding for Controlled Text Generation"

* The discussion on ties ends up being unecessary, as it does not impact performance. Thus, you might not want to state "Accounting for ties in reward modeling is especially important" (l.243).

**Questions:**

See weaknesses.

---

> ### Author Response · Authors · 2024-11-23
> **Response to Reviewer Pqum**
>
> We thank the reviewer for their comments and for engaging with the paper. To address the concerns, we add a more complex reasoning domain, Math 500 and add additional clarity to the computational efficiency of our algorithms with the measure of the number of FLOPs. We are happy to clarify any questions you may have. Please let us know if your concerns are addressed and if so we would be grateful if you would be willing to raise your score. We would be happy to discuss if you have any concerns
>
> **Q1:** Are the improvements over baselines small?
>
> **A1:** Thank you for your question. Please see the sections “What are we improving?” and “How much are we improving it by?” in the overall response.
>
> In brief, self-evaluations and early pruning reduce costs (computation, memory, and energy consumption) by a factor of 4 without introducing additional latency. Adaptive sampling further reduces costs by approximately a factor of 6 to 8, though it comes with the tradeoff of increased latency. Additional details on these measurements are included in the overall response.
>
> **Q2:** Why doesn't the paper mention process-based reward models?
>
> **A2:** Thank you for bringing this up and for the helpful reference! Note, we do not intend to improve the performance of the given inference-time method, which we fix in this work to be Best-of-N. We simply aim to make it more cost-effective by removing the need for an external reward model as well as adaptively reducing the amount of computation needed depending on the query.
>
> In this work, we learn reward models with an on-policy preference dataset, utilizing prompts from a large open-source general dataset, LMSYS. In contrast, PRMs in the context of reasoning require the need for ground truth outcome supervision. For domains such as Math and Code, supervision can be provided by symbolically checking the final answer in the context of math or executing test cases in code. A standard paradigm explored in works such as Snell et al, 2024 and OmegaPRM (Luo et al 2024) is to learn a PRM through MC rollouts. These are two complementary approaches for training reward models, with different assumptions and different dataset compositions. However, the inference time optimizations we propose such as adaptive sampling + pruning, could be readily applied to any reward model, allowing for savings in FLOPs during inference. We will try to incorporate a PRM for the final version of the paper to showcase the efficacy of this approach. We will additionally add these works to the related works section of the paper and will map out how our method can be extended to other inference-time methods very naturally.
>
> **Q3:** How does the need to generate sequentially affect real-time deployment, and can you address this limitation more thoroughly? Is the idea of exponentially increased batch size sufficient to mitigate this issue?
>
> **A3:** Thank you for raising this point. Please refer to the sections “What are we improving?” and “How much are we improving it by?” in the overall response.
>
> In summary, self-evaluations and early pruning reduce costs (computation, memory, and energy consumption) by a factor of 4 without additional latency. Adaptive sampling further reduces costs by a factor of 6 to 8 but introduces approximately 2x additional latency, which can be mitigated using exponentially increasing batch sizes. More details on these measurements are available in the overall response.
>
> **Q4:** Can you discuss how your training method effectively trains the model to estimate the quantile of the generation according to the external reward model?
>
> **A4:** If $\epsilon = 0$ in the construction of the preference dataset (i.e ties only if the response is identical), capability aware self evaluation reduces to estimating the quantile of the given response. However, practically this is difficult to model (the Best-of-N performance suffers), which is one of the reasons we introduced a non-zero epsilon as a relaxation of this interpretation. Intuitively, it is easier to predict with high confidence if the model can’t do significantly better rather than if the model can’t do better by any amount. This relaxation is fine as we do not strictly care about the quantile of a response, but instead if a new response could be significantly better than the current generated response. This is the reason our model can routinely make confident predictions such as  P(Win or Tie) = 0.99, indicating that the probability of a significant improvement is very low.

---

> ### Author Response · Authors · 2024-11-23
> **Response to Reviewer Pqum (Continued)**
>
> **Q5:** Could you consider extending the method to learn to detect whether a given generation is the best among N>2 generations?
>
> **A5:** Absolutely! Preference datasets usually only have two responses per query but this is a great suggestion, we could definitely extend this method to consider the probability that N>2 number of generations will result in at least one better generation. This is an interesting way of augmenting the dataset, as well as producing potentially more useful predictions to more efficiently allocate samples to a given query. We can try to incorporate this for the final version of the paper and will add this to the discussion section of our paper for completion.
>
> **Q6:** Can you provide ablation studies, such as applying temperature annealing to basic Best-of-N?
>
> **A6:** Thank you for the suggestion. We find that our annealing schedule method (annealing based on samples generated so far) seems to provide gains even without adaptive sampling at small Best-of-Ns (+2% on GSM8K for Best-of-2). However, benefits taper off at larger Best-of-Ns. This makes sense as our scheduling strategy only applies significant annealing to the first few samples. We will introduce this and additional ablations for the final revision of the paper.
>
> **Q7:** Could you explore a simple strategy that detects prompt complexity and allocates compute accordingly?
>
> **A7:** This is a great suggestion. However, practically, we had found limited success with this approach as estimating the prompt complexity is quite difficult to use to then predict the required amount of compute given only the query. This is what motivated the experiments with mid-generation self-evaluation (early pruning), allowing the model to perform a limited amount of inference (e.g 64 or 128 tokens) and then prune according to the complexity of the problem. We will include this as a formal baseline for the final revision of the paper.
>
> **Q8:** Could you clarify why you need the model's prior distribution over tokens to be similar after training to keep the underlying policy unchanged? Isn't there a risk of catastrophic forgetting when fine-tuning for self-evaluation?
>
> **A8:** Thank you for bringing this up! This is a small but important detail in practice. We need the model’s distribution over tokens to be similar before training, as this will minimize the amount the model needs to change to minimize the loss during training. This could be thought of as reducing the risk of catastrophic forgetting, but more concretely, we do not want the model to overgeneralize and learn to only output the target tokens (“Yes” and “No”). We found that If the self-evaluation prompt elicits these tokens by default (this should be verified before training), the model does not overgeneralize and we effectively avoid this problem. One way to check if this was done correctly is to make sure that the loss is already reasonable before fine-tuning.

---

> > ### Comment · Reviewer_Pqum · 2024-11-26
> >
> > I would like to thank the authors for their rebutal. However
> > - I don’t think you can say that your method « reduces by a factor of 4 without introducing additional latency » as you actually require sequential generations while the standard BoN can be distributed.
> > - « We will try to incorporate a PRM for the final version of the paper to showcase the efficacy of this approach »: I believe this baseline is important  to justify the paper title « LLMs Can Predict if They Can Do Better, Even Mid-Generation » .
> >
> > Therefore, I will keep my score (weak reject) but will be open to debate if there is a strong opinion from other reviewers. Best.

---

> ### Author Response · Authors · 2024-11-27
> **Response to Reviewer Pqum**
>
> We thank the reviewer for their continued engagement!
>
> > You require sequential generations while the standard BoN can be distributed
>
> To clarify, we propose two approaches: **early pruning** and **adaptive sampling**. Early pruning does not require any sequential processing of samples, while adaptive sampling does. Specifically:
>
> 1. **Early Pruning**: This approach reduces FLOPs by **4x** to match the accuracy of Best-of-16 with **no additional latency**. It reduces computational overhead by halting the generation of less promising samples **during parallel generation**. For example, after generating 128 tokens for 16 samples in parallel, the 12 least promising samples can be pruned, effectively halving the total number of tokens generated and achieving significant computational savings.
>
> 2. **Adaptive Sampling**: This approach achieves a larger reduction in FLOPs by **6-8x** to match the accuracy of Best-of-16, albeit with additional latency due to sequential generations. However, the average latency can be as low as 1.1 sequential generations, thanks to further optimizations such as the exponentially annealed temperature schedule, that we propose, resulting in limited additional overhead over a batch of questions.
>
> > PRM as a baseline
>
> We note that the Process Reward Model (PRM) can readily be replaced with the self-evaluation function we propose. The inference speedups we present (adaptive sampling + pruning) remain valid and transferable with this function.
>
> The key difference between a PRM and our mid-generation self-evaluation approach lies in the dataset composition used to train these functions. PRMs (e.g., Snell et al., 2024; Wang et al., 2024, Math Shepherd) are trained on policy rollouts evaluated for correctness via outcome verification. In contrast, our self-evaluation frameworks collects the same on-policy rollouts but utilizes relative preference feedback instead of absolute feedback. This decision to train on general queries from preference data (e.g., LMSYS) enables our single model to generalize across a wide range of domains.
>
> In many domains, learning an accurate PRM is challenging due to the need for highly accurate outcome supervision models. Additionally, for some domains such as creative writing or general question answering, such supervision can be ambiguous and difficult to collect effectively. Moreover, inference with an external PRM function is computationally expensive, requiring additional FLOPs and memory, particularly for mid-generation execution.
>
> Nevertheless, for the rebuttal/final version of the paper, we will attempt to train a PRM for math contest problems using domain-specific datasets, such as PRM800K or Math Shepherd. Additionally, we are open to rephrasing the title to temper any overstated claims.

---

> > ### Comment · Reviewer_Pqum · 2024-11-27
> >
> > Thank you the additional comment;
> >
> > "learning an accurate PRM is challenging" -> actually I believe that taking a given RM trained on offline dataset, and then finetuning it on the online mig-generation would be a competitive and simple baseline. Without this ablation, I am afraid we lack real information about the practicality of using the LLM to self evaluate.

---

> > > ### Author Response · Authors · 2024-11-29
> > > **Adding PRM Experiment**
> > >
> > > This is a very helpful suggestion! This allows us to compare our method to a strong value model instead of a domain-specific PRM. Following what you described, we finetuned ArmoRM (the same reward model we used to create our preference data) to predict the reward given truncated generations. This is a very strong baseline, as ArmoRM was already trained on 1 million preferences and we further finetuned it to predict the reward directly instead of having to predict $P(Win \cup Tie)$ as our model is trained to do. Compared to Best of N, we found that this baseline does allows for some saving in FLOPs, with the pruning of less desirable responses. However, this gain is completely overshadowed by the additional cost of having to process all tokens a second time due to the lack of KV cache. This results in our method still saving almost **2x** more FLOPs. You can find an updated performance vs FLOPs graph **[here](https://imgur.com/a/pdQ878w)**. We will add these results to our paper in the final revision. We would be most grateful if you would consider upgrading your score, given that your concerns have now been addressed.

---

> > > > ### Comment · Reviewer_Pqum · 2024-12-02
> > > >
> > > > I would like to thank the authors for this interesting experiment. But as Reviewer PBQS mentionned, "I am now more confused about what is the contribution of this work. The contribution now seems a set of trick to improve the efficiency at inference-time.".  Overall, I believe the paper's contributions lack clarity. Thus, I will not change my overall grade 5, but as mentionned earlier, I will not block the paper.

---

> > > > > ### Author Response · Authors · 2024-12-03
> > > > > **Response to Reviewer Pqum**
> > > > >
> > > > > Thank you for your continued engagement and feedback. To address your concerns about the nature of our contributions, we encourage you to refer to the newly added section "How are we improving it?" in the revised "Overall Response." This section clearly outlines our approach, distinguishing our key contributions.
> > > > >
> > > > > We sincerely thank you for your time during this discussion period.

---

### Official Review · Reviewer_6jZH · 2024-11-01

**Soundness:** 3
**Presentation:** 3
**Contribution:** 2
**Rating:** 6
**Confidence:** 4

**Summary:**

The paper shows that one of the challenging problem at inference-time computation scaling (Best-of-N) is to identify if a specific generation is "good", or if it is worth allocating more compute to continue its generation. For example, Best-of-N is considered computationally expensive because it requires external reward model to evaluate the quality of the generation, and it often requires multiple such generation samples to obtain a good quality result.

To address this problem, the paper introduces capability-aware self-evaluation, a technique that allows LLM to self-predict if the current generation is promising, or in other words, if restarting the generation will yield better result or not. In the Best-of-N setting, the paper also introduces adaptive sampling and annealing to further improve the quality of the generation, and early prune unpromising samples using the self-evaluation methodology.

In the evaluation section, the paper presents a finetuned Llama3.1-8B-Instruct model trained on 30k preference pairs constructed from unfiltered LMSYS using a reward model, and shows an increasing win rate against GPT-4.

**Strengths:**

**Elegant solution**. The evaluation section shows that the proposed fine-tuning method is able to generalize from LMSys to AlpacaEval and GSM8K. The solution is elegant and intuitive, and it works well.

The method shows a promising way to improve the quality of generation in Best-of-N setting by early pruning unpromising samples, and adaptively sampling and annealing the remaining samples. The proposed method is simple, intuitive, and shown to be effective.

**Writing**. The writing is clear and the paper is well-organized. The paper is well-written and easy to follow.

**Weaknesses:**

**Finetuning dataset construction - what is a good dataset to finetune on?**. Why is LMSYS dataset chosen? Would another dataset other than LMSYS do better or worse? Would using a more complex dataset do better or worse? It is not as convincing if there is no ablation study on the dataset choice.

**Limited evaluation**. Only 2 datasets are used to perform the evaluation. GSM8K is not a hard dataset to see performance gain (e.g. using hyperparam tuning). To show the generalizability on reasoning tasks, it would be better to evaluate on more diverse and complex datasets, especially on slightly longer reasoning chains to show the increasing size of sampling and prunning is effective. Showing the distribution of generated token lengths would be very useful to understand "how far" are we talking about.

**Limited setting of inference time algorithm**. As mentioned in conclusion / limitation section, the paper only discuss Best-of-N setting. But many reasoning techniques such as self-refinement / chain-of-thought are not generating parallel requests. It would be better to evaluate on more diverse inference settings.

**Naively mapping number of samples / tokens to computational cost**. The paper did not include any statement about computational cost in terms of number of tokens generated. In actual LLM serving with batch inference (e.g. vLLM, DeepSpeed), the computational cost of 16 samples performing decoding is roughly the same as 1 sample performing decoding. Thus, if the metric is "end-to-end latency", then the bottleneck is actually the straggler - meaning what is the longest request in the batch that needs to be finished before the generation is complete. The proposed method does not seem to address this problem. In addition, as mentioned in discussion section, introducing a prefill (inserting prompt into the middle of generation) is not free, and may actually introduce more latency than naive batch decoding setting. Although this is not the focus of the paper, it is important to properly address this point in writing as it is not very clear at reading.

**Questions:**

1. Does the selection of finetuning datasets matters, or to what extent it affect the generalization of the model?
2. What is the distribution of the generation length?
3. Would the proposed method still be effective when problem is more difficult or unseen?

---

> ### Author Response · Authors · 2024-11-23
> **Response to Reviewer 6jZH**
>
> We thank the reviewer for their comments and for engaging with the paper. To address the concerns, we add a more complex reasoning domain, Math 500 and add additional clarity to the computational efficiency of our algorithms with the measure of the number of FLOPs. We are happy to clarify any questions you may have. Please let us know if your concerns are addressed and if so we would be grateful if you would be willing to raise your score. We would be happy to discuss if you have any concerns.
>
> **Q1:** Why did you choose the LMSYS dataset for fine-tuning, and how does the selection of the fine-tuning dataset affect the generalization of the model? Would using a different or more complex dataset improve performance?
>
> **A1:** Thank you for your question. We selected the LMSYS dataset for fine-tuning because it consists of real-world prompts. Our goal was to enable the model to adaptively allocate samples across a wide range of domains, and the broad distribution of prompts in LMSYS supports this objective.
>
> We evaluated the fine-tuned model on diverse domains, including AlpacaEval, GSM8K, and now Math 500, which demonstrates its generalization capabilities. Since our performance already closely approaches the theoretical upper bound set by the underlying reward model used in constructing the preference dataset, we believe that training on a different dataset would likely yield minimal improvement. However, fine-tuning on more specific, complex datasets could enhance performance on similar in-distribution tasks, and this is something we may explore in a future revision.
>
> **Q2:** Can you evaluate your method on more diverse and complex datasets, especially those requiring longer reasoning chains, to demonstrate its generalizability on reasoning tasks?
>
> **A2:** To address this concern, we added the Math 500 (Hendrycks et al, 2020) as an additional domain as seen in our new figure which can be found [here](https://imgur.com/a/rhisaLK). This domain is more complex, requiring longer reasoning chains to perform harder math contest problems. We find that the results are relatively consistent with what we see on GSM8K.
>
> **Q3:** Can you show the distribution of generated token lengths?
>
> **A3:** Certainly. Below are the token length statistics for the evaluated datasets:
>
> For AlpacaEval 2.0, the mean response length is 453 tokens with a standard deviation of 305 tokens. For GSM8K, the mean response length is 251 with a standard deviation of 170 tokens. For MATH 500, the mean response length is 537 tokens with standard deviation of 337 tokens. We will add these metrics and a histogram of the generated tokens to our appendix by the end of the rebuttal period or for the final version of the paper.
>
> **Q4:** The paper maps the number of samples or tokens directly to computational cost, but in practice, batch inference can make multiple samples cost similar to one sample. Can you address how your method impacts computational cost and latency in real-world LLM serving scenarios?
>
> **A4:** Thank you for this important question. Please refer to the sections “What are we improving?” and “How much are we improving it by?” in the overall response. In those sections, we provide concrete metrics and explanations on the cost-effectiveness of our approach, including its impact on computational cost in practical settings.
>
> **Q5:** Considering that inserting a prefill (self-evaluation prompt) is not free and may introduce more latency than naive batch decoding, can you clarify how your method affects end-to-end latency?
>
> **A5:** We appreciate your question. Please see the section “How much are we improving it by?” in the overall response. There, we provide concrete metrics and a detailed explanation of the tradeoff between the cost of a reward model and the self-evaluation process, as well as their respective impacts on end-to-end latency.

---

> > ### Comment · Reviewer_6jZH · 2024-11-26
> > **Thank you**
> >
> > Thank you for your detailed comment. I think the reply addresses my comments.

---

> > > ### Author Response · Authors · 2024-11-27
> > >
> > > We would like to express our sincere thanks and appreciation for your feedback. We would be most grateful if you would consider upgrading your score, given that your concerns have now been addressed.

---

### Official Review · Reviewer_PBQS · 2024-11-02

**Soundness:** 2
**Presentation:** 2
**Contribution:** 2
**Rating:** 3
**Confidence:** 3

**Summary:**

A method for predicting whether the answer to a query being generated by a LLM is worth continuing or not.
The method fine-tunes a pre-trained LLM to predict whether the current (partially) generated answer is worth   continuing or not. The fine-tuning involves pre-pending the prompt 'Would you do better if you started over?   (“Yes.” or “No.”)' to the answer of a query and SFT the model to predict "Yes" if the answer is prefered over  the alternative answer.
By deciding to abort the answer generation, the authors claim that the LLM may save compute at inference-time  compared to a method that do not abort the answer generation.

**Strengths:**

The paper tackles an important challenge and chose to do so using an open-weight model. Advances on open-      weight models may benefit everyone and reducing the compute and inference time may increase the adoption of    foundation models. The method itself was understandable (althought I believe that the writing of the paper     could generally be made clearer and more succinct).

**Weaknesses:**

The paper proposes a method that improves inference-time compute, yet it never compares the compute-time and   the FLOPs with baseline methods. Having to probe the model mid generation definitely incurs a cost, which is   not being discussed in the paper. Moreover, the model finds that pruning longer sentence improves performance, in which case perhaps it would be better to fully generate the sentences before selecting them and thus the    idea of pruning mid-generation is unclear.

In Figure 1, the x-axis stops at 16 Fully Generated Samples, where the method of the authors has hit a plateau and the Best-of-N method is still showing improvement. The authors should increase the amount of fully         generated samples to observe and compare the effect of more generated sample of their method and the Best-of-N baseline.

In Table 1 the author compares baseline methods with the proposed method. However, it is not at all clear what are the parameters of the capability-aware method and whether it is comparable to the baselines. The authors   have to clarify how they compare with the baseline methods. E.g. of questions I have (and this is a subset of  the questions I have -- the authors should make sure that the revised version clearly explain the experimental methodology): How many fully generated samples are used? (Line 404 the authors mention Best-of-16 sampling, is this 16 fully generated samples?) If this is indeed 16 fully generated samples, then why should a pratitioner  use that method instead of the best-of-N baseline which performs comparably well acccording to Figure 1.

The ablations that are important for understanding the parameters of the method are in the Appendix while they should be in the main paper.

The authors continually refer to Figure 1 throughout the paper, forcing the reader to continually scroll up    and down. Instead of packing many important observations within one figure, the authors are encouraged to      break down the observations into indivudal figure (i.e. one observation per figure).

Finally, I encourage the authors to revise the writing making it more succinct and avoid long winded sentences.

**Questions:**

* How does the proposed method compares with the baselines in term of compute-time?
* How does the proposed method compares iwth the baselines in term of FLOPs?
* How does Figure 1 looks like when evaluating 32 and 64 fully generated samples?
* What are the parameters of the methods and the baselines used to produce table 1?

---

> ### Author Response · Authors · 2024-11-23
> **Response to Reviewer PBQS**
>
> We sincerely thank the reviewer for their comments and for engaging with the paper. To address the concerns, we add a more complex reasoning domain, Math 500 and add additional clarity to the computational efficiency of our algorithms with the measure of the number of FLOPs. We are happy to clarify any questions you may have. Please let us know if your concerns are addressed and if so we would be grateful if you would be willing to raise your score. We would be happy to discuss if you have any concerns.
>
> **Q1:** How does the proposed method compare with the baselines in terms of compute time and FLOPs, considering the cost of probing the model mid-generation?
>
> **A1:** Thank you for this question. Please see the sections “What are we improving?” and “How much are we improving it by?” in the overall response for detailed insights.
>
> In brief, self-evaluations and early pruning reduce costs (computation, memory, and energy consumption) by a factor of 4, without introducing any additional latency. Adaptive sampling further reduces costs by approximately a factor of 6 to 8, though it comes with the tradeoff of additional latency. Additional details on how these improvements are measured are included in the overall response.
>
> **Q2:** Since pruning longer sentences improves performance, wouldn't it be better to fully generate sentences before selecting them? Why is pruning mid-generation necessary?
>
> **A2:** This is a good question. Please refer to the sections “What are we improving?” and “How much are we improving it by?” in the overall response.
>
> To clarify, our approach does not aim to improve the maximum theoretical performance of Best-of-N but rather to make Best-of-N far more cost-effective. Specifically, early pruning with self-evaluations reduces costs by a factor of 4. Generating a larger number of responses and pruning some early—rather than generating fewer full responses—is shown to be computationally more efficient. This method can be viewed as improving downstream task performance under fixed compute, memory, or energy budgets. The details of these measurements can be found in the overall response.
>
> **Q3:** Why does your method hit a plateau while the Best-of-N method still shows improvement over more samples? Would Best-of-N surpass your method when evaluated at 32 and 64 fully generated samples?
>
> **A3:** Thank you for raising this point. Please refer to the section “What are we improving?” in the overall response for more details.
>
> As shown in Figure 1, we plot accuracy against the number of samples, with a fixed maximum sample budget. Best-of-16 serves as the oracle or upper-bound performance for a given reward model. Our methods, such as Adaptive Sampling and Pruning, do not increase the theoretical maximum performance of Best-of-N or other search algorithms but instead significantly reduce the costs associated with achieving that performance (in terms of computation, memory, and energy consumption). We will evaluate the performance at 32 and 64 samples and will include these results either by the end of the rebuttal period or in the final version of the paper.
>
> **Q4:** Can you move important ablation studies from the Appendix to the main paper for better understanding of your method's parameters? Can you restructure your figures to present one observation per figure instead of combining many into Figure 1 to improve readability? Could you revise the writing to make it more succinct and avoid long, complex sentences?
>
> **A4:**  Thank you for these suggestions. We will take the following steps to improve clarity and readability in the final revision of the paper:
>
> 1. We will add clarity as you have suggested by reducing text and complex sentences in the methods section of the paper.
> 2. Additionally, we will move Figure 1 to the experiments section of the paper and break it into two separate figures to improve readability.
> 3. Finally, we will move key Tables and Figures from the appendix to give better understanding of the method's parameters.
>
> We would be happy to make further changes that lead to a better understanding of this paper.

---

> > ### Author Response · Authors · 2024-11-27
> > **Following up!**
> >
> > Thank you for your review! Please let us know if further detail is needed or if the new experiments address your concerns.

---

> > > ### Comment · Reviewer_PBQS · 2024-11-28
> > > **Response**
> > >
> > > I appreciate that you found the feedbacks from the reviewers helpful. I read through all of the reviews and response which clarified some of my questions and concerns. However, severe issues still remain:
> > > * While the authors provide a comparison of the FLOPs in their rebuttal, this comparison is not discussed nor presented in the paper.
> > > * The authors still do not provide a comparison of the compute time (at least not in a format that I can understand) between the baselines. Considering the angle of this paper and given the remark of pws6, this is an important metric to highlight.
> > > * The authors could modify their paper during the rebuttal period to improve its clarify, yet they have not modified it and thus my concerns about clarity still remain in the current version.
> > > * After reading the responses, I am now more confused about what is the contribution of this work. The contribution now seems a set of trick to improve the efficiency at inference-time. In which case, a careful ablation has to be made so that the reader understand the impact of each tricks.
> > > * There are competitive methods out there, as highlighted by Pqum, that have not been considered nor compared against in this paper. Such comparison is important for the type of paper presented here.
> > >
> > > While I appreciate the effort put by the authors in the rebuttal, I believe that more work is needed to help the reader understand the impact of each tricks proposed in the paper and comparison against strong baseline, or using the strongest baseline in the literature as starting point, is important.
> > >
> > > Finally,I will update my confidence score from a 4 to a 3.

---

> > > > ### Author Response · Authors · 2024-11-28
> > > > **Response to PBQS**
> > > >
> > > > We appreciate the reviewer’s continued engagement and constructive feedback on our work. We are pleased to share an updated version of the paper, addressing the questions and concerns raised.
> > > >
> > > > > While the authors provide a comparison of the FLOPs in their rebuttal, this comparison is not discussed nor presented in the paper. [...]The authors could modify their paper during the rebuttal period to improve its clarify, yet they have not modified it and thus my concerns about clarity still remain in the current version.
> > > >
> > > > We have now revised the paper to include the comparison of FLOPs in the main text. Specifically, we have restructured the writing and updated the figures to emphasize this as the primary form of comparison, aligning with prior works in inference optimization (e.g., Hoffman et al., *Training Compute-Optimal Large Language Models*). Additionally, we have improved the clarity of the experimental section, incorporating the reviewer’s suggestion to relocate Figure 1 (or equivalent) to this section. To further enhance presentation, we have fragmented Figure 1 which previously encompassed both pruning and adaptive sampling into two separate figures. We have additionally modified the writing of the experimental section to reflect this.
> > > >
> > > > We welcome any additional formatting or content suggestions and are prepared to share an anonymous PDF reflecting any changes requested by the reviewer for the final version of the paper.
> > > >
> > > > > Regarding PRMs
> > > >
> > > > The Process Reward Model (PRM) can be seamlessly replaced with our proposed self-evaluation function. The inference speedups presented in the paper—adaptive sampling and pruning—remain valid and transferable with this approach.
> > > >
> > > > The distinction between PRMs and our mid-generation self-evaluation lies primarily in the dataset composition used for training. PRMs (e.g., Snell et al., 2024; Wang et al., 2024, Math Shepherd) rely on policy rollouts evaluated for correctness using outcome verification. By contrast, our self-evaluation framework employs relative preference feedback from the same on-policy rollouts. This choice, informed by preference data (e.g., LMSYS), enables our model to generalize across diverse domains with a single framework.
> > > >
> > > > We are actively working on adding a PRM baseline for comparison before the end of the rebuttal period and will include these results in the revised version.
> > > >
> > > >
> > > > > The authors still do not provide a comparison of the compute time (at least not in a format that I can understand) between the baselines. Considering the angle of this paper and given the remark of pws6, this is an important metric to highlight.
> > > >
> > > > In the paper, we study latency as the average number of batches or sequential calls to the language model that are made. This is an **overestimate** of the latency in many applications due to optimizations such as prefix-caching and batched generation.
> > > >
> > > > We now include wall-time as an additional metric for measuring latency. We evaluate wall-time in a controlled environment using SGLang (Zheng et al.) on a consistent hardware setup (8×A100 GPUs), reporting the average wall-time over questions. Figure 3(B) in the updated paper presents wall-clock time (in seconds) versus the percentage of maximum improvement. These results demonstrate that pruning incurs no additional latency even with a 4× increase in FLOPs, while adaptive sampling introduces additional latency that can be balanced with performance gains and FLOPs.
> > > >
> > > > > After reading the responses, I am now more confused about what is the contribution of this work. The contribution now seems a set of trick to improve the efficiency at inference-time. In which case, a careful ablation has to be made so that the reader understand the impact of each tricks.
> > > >
> > > > We believe the contributions of our work are well-supported through extensive ablation studies, both in the Appendix and the main paper. First, Table 2 examines the effect of modeling probability for self-evaluation on an on-policy pairwise preference dataset. Next, in Table 1, we study different instantiations of reward modeling that can be used for guiding inference time search. For adaptive sampling, Table 4 assesses the effect of omitting the annealing schedule, Table 5, compares using a parametric reward model. For pruning, we study randomly pruning, pruning at a fixed token budget, and no pruning in Tables 7-9.
> > > >
> > > > We are open to conducting additional ablation studies if the reviewer deems further analysis necessary. However, we believe the current ablations provide a comprehensive evaluation of the impact of each component in our proposed framework.

---

> ### Author Response · Authors · 2024-11-29
>
> > There are competitive methods out there, as highlighted by Pqum, that have not been considered nor compared against in this paper. Such comparison is important for the type of paper presented here.
>
> Thank you for this comment. We now include the baseline the reviewer suggested to the paper. We copy the response below for your convenience:
>
> This is a very helpful suggestion! This allows us to compare our method to a strong value model instead of a domain-specific PRM. Following what you described, we finetuned ArmoRM (the same reward model we used to create our preference data) to predict the reward given truncated generations. This is a very strong baseline, as ArmoRM was already trained on 1 million preferences and we further finetuned it to predict the reward directly instead of having to predict $P(Win \cup Tie)$ as our model is trained to do. Compared to Best of N, we found that this baseline does allows for some saving in FLOPs, with the pruning of less desirable responses. However, this gain is completely overshadowed by the additional cost of having to process all tokens a second time due to the lack of KV cache. This results in our method still saving almost **2x** more FLOPs. You can find an updated performance vs FLOPs graph **[here](https://imgur.com/a/pdQ878w)**. We will add these results to our paper in the final revision.
>
> We would be most grateful if you would consider upgrading your score, given that your concerns have now been addressed.

---

### Official Review · Reviewer_pws6 · 2024-11-04

**Soundness:** 2
**Presentation:** 3
**Contribution:** 2
**Rating:** 6
**Confidence:** 4

**Summary:**

- The paper proposes to simple way to train a LLM to do self-evaluation (with additional tokens) and uses the logits for the trained additional tokens to decide whether to continue sampling or not. They show that they can approximately match the performance of Best-of-16 at, on average, 1/4 of the number of samples on GSM8K.
- The authors claim on the efficiency and latency of the proposed method, but since it requires serial, instead of parallel, generations when doing Best-of-N, the latency may actually increase by a non-negligible amount. This needs to be properly addressed in the paper as this is a paper about making Best-of-N efficient.

**Strengths:**

- The authors propose a very simple and concrete way to adaptively scale generations for Best-of-N and show an noticeable improvement on GSM8K where they can approximately match the performance of Best-of-16 at 1/4 of the number of samples.

**Weaknesses:**

- The performance gain is marginal for Alpaca Eval. Thus, more comprehensive evaluations on a broader range of benchmarks that perhaps require more complex reasoning will be beneficial. However, I fully acknowledge the limits of what can be done during the rebuttal.
- The paper/authors argue that their proposed adaptive inference-time compute is scalable and efficient based on the performance vs. number of samples comparison. They make an argument on latency. However, an important aspect that is underplayed is the actual computational efficiency when implemented. Although mentioned, because of the need to self-evaluate, the adaptive inference-time compute is allocated in serial. On the other hand, Best-of-N can be executed in parallel, which is its advantage over this method. Although inference speed is very dependent on the inference stack engine, it would be important to quantify this cost of serial processing.
- Some further analysis is warranted to analyze the effect of learning to self-evaluate alone. Interestingly, the authors note that a pretrained (and instruction tuned) model has a nontrivial ability to conduct self-evaluations and that a Bradley Terry reward model further improves it. It would be constructive to study the effect of self-evaluation training in isolation and essentially conduct Best-of-N after this self-evaluation training to separate the effect of this training on newly annotated data and the adaptive inference method.

**Questions:**

- For Section 4.2 experiments on AlpacaEval and GSM8K, what is the performance of Llama 3.1 8B Instruct when using Best-of-16?
- The authors focus on showing the efficiency in terms of the reduction of the number of samples vs performance. However, an interesting experiment would be to, fix the amount of compute budget (e.g., FLOPS), and to see whether one can adaptively allocate higher # of samples for more difficult questions and fewer for easier questions, while using everything. That would essentially show the effectiveness of this adaptive compute allocation method compared to a naive Best-of-N that allocates the same amount of compute to any prompt.
- How does the proportion of ties (the epsilon for deciding ties) in the preference dataset construction for self-evaluation training change the performance of the model?
- I am not sure how realistic this is given the time constraint of the rebuttal, but a natural question that arises is how this changes for datasets that require more complex reasoning such as ArenaHard, Math Hard, etc.

---

> ### Author Response · Authors · 2024-11-23
> **Response to Reviewer pws6**
>
> We thank the reviewer for their comments and for engaging with the paper. To address the concerns, we add a more complex reasoning domain, Math 500 and add additional clarity to the computational efficiency of our algorithms with the measure of the number of FLOPs. We are happy to clarify any questions you may have. Please let us know if your concerns are addressed and if so we would be grateful if you would be willing to raise your score. We would be happy to discuss if you have any concerns.
>
> **Q1:** Are the performance gains marginal for Alpaca Eval?
>
> **A1:** Thank you for your question. Please refer to the sections “What are we improving?” and “How much are we improving it by?” in the overall response for a detailed explanation.
>
> Specifically, for AlpacaEval, we employed self-evaluations and early pruning to significantly reduce the cost (in terms of computation, memory, and energy consumption) of Best-of-16 by approximately a factor of 4. Additional details on how these improvements are measured can be found in the overall response.
>
> **Q2:** Can you quantify the actual computational efficiency and latency of your methods compared to Best-of-N, considering that your method requires serial processing while Best-of-N can be executed in parallel? How does your method perform on datasets that require more complex reasoning?
>
> **A2:** Thank you for this suggestion. The section “How much are we improving it by?” in the overall response contains the necessary details that we will add to our final revision.
>
> In summary, self-evaluations and early pruning reduce costs by roughly a factor of 4 without incurring additional latency. Meanwhile, adaptive sampling achieves cost reductions of approximately 6 to 8 times, though it introduces some additional latency as a tradeoff. Further details about these measurements can be found in the overall response.
>
> Regarding complex reasoning tasks, we evaluated the method on MATH 500, a much more challenging dataset, and observed results consistent with GSM8K. This evaluation is also discussed in the “How much are we improving it by?” section.
>
> **Q3:** Can you study the effect of self-evaluation training in isolation by conducting Best-of-N after self-evaluation training to separate the effect of this training from the adaptive inference method?
>
> **A3:** To observe the effect of self-evaluation training, please refer to Table 1. The table highlights the difference in performance between zero-shot (LLM-as-a-Judge), reward modeling (Bradley-Terry), and our capability-aware self-evaluations. To clarify, here we utilize the same samples from Llama 3.1 8B-Instruct per question and evaluate solely the reward modeling/self-evaluation capabilities of the model. Here, we show that Best-of-16 performance improves from 24.4% to 33.8% on AlpacaEval and improves from 86.7% to 91% on GSM8K when comparing LLM-as-a-Judge to capability-aware self-evaluation. Additionally, we see 33.2% to 33.8% and 87.7% to 91% in the same domains when comparing a Bradley Terry reward model to capability-aware self-evaluation. In short, the self-evaluation training allows us to provide a significant performance benefit over traditional Best-of-N with a reward model or using the zero-shot ability of the model in the LLM as a Judge formulation.
>
> **Q4:** What is the performance of Llama 3.1 8B Instruct when using Best-of-16 on AlpacaEval and GSM8K?
>
> **A4:** As shown in Table 1, with Llama 3.1 8b Instruct, capability-aware self-evaluation gets a Best-of-16 performance of 33.8% on Best-of-16 on AlpacaEval and 91% on Best-of-16 with GSM8K. For the learned reward model, the Best-of-16 on AlpacaEval is 33.2% and the Best-of-16 with GSM8K is 87.7%.
>
> **Q5:** How does the proportion of ties (the epsilon for deciding ties) in the preference dataset construction for self-evaluation training affect the performance of the model?
>
> **A5:** This is an excellent question. Many responses in an on-policy preference dataset will be semantically similar as they come from the same model as seen in prior work (Tajwar et al, 2024). Therefore, relaxing the Bradley-Terry Model to account for ties can make the prediction problem easier. Intuitively, with a large proportion of ties, we increase the semantic distance between the winning and losing response, making self-evaluation easier. This is the reason our model can routinely make confident predictions such as  P(Win or Tie) = 0.99, indicating that the probability of a significant improvement is very low.

---

> > ### Comment · Reviewer_pws6 · 2024-11-25
> > **Reviewer Response**
> >
> > Thank you for the detailed responses and the overall clarification of the contributions of this work.
> >
> > Having read the responses from the authors, it is now my understanding that the main contribution of this work is to propose a self-evaluation method for early pruning and adaptive sampling. The trade-off of adaptive sampling is that it introduces additional latency due to serial processing. Although very helpful, it seems that FLOPS, although a common metric for efficiency, may only highlight the strength in terms of the amount of computation necessary, but does not accurately capture the latency aspect.
> >
> > The authors also mention that they test on MATH 500, but would you please point me to where I can find the new experimental results?
> >
> > Based on this discussion, I will raise my score to a 6.

---

> > > ### Author Response · Authors · 2024-11-25
> > > **Response to Reviewer pws6 [1/1]**
> > >
> > > We appreciate the reviewer’s thoughtful comments and continued engagement in the discussion.
> > >
> > > > The trade-off of adaptive sampling is that it introduces additional latency due to serial processing
> > >
> > > While adaptive sampling indeed introduces additional latency, we also propose an alternative solution, early pruning, a technique that incurs *no additional latency* while improving FLOPs efficiency and reducing the number of tokens generated. For latency-sensitive settings, early pruning may serve as an attractive alternative.
> > >
> > > > FLOPS, although a common metric for efficiency, may only highlight the strength in terms of the amount of computation necessary
> > >
> > > This is an excellent point. FLOPs primarily reflect the computational requirements. To address this, we also report the average batches per query (for sequential queries) in our results, such as those in Table 3 (reproduced below for reference). Our method incorporates exponentially increasing batch sizes to minimize latency for challenging queries, enabling more efficient allocation of inference time compute. Additionally, we now propose warm-starting with an average batch size (rounded) for a desirable target accuracy, further reducing latency for subsequent queries.
> > >
> > > | Win-or-Tie Probability Threshold | 0.92 | 0.96 | 0.98 | 0.99 | 1.00 |
> > > |----------------------------------|------|------|------|------|------|
> > > | Average Samples Used             | 1.2  | 1.9  | 3.7  | 6.8  | 16.0 |
> > > | Average Batches Used (Latency)   | 1.1  | 1.4  | 2.0  | 2.9  | 5.0  |
> > > | GSM8K Pass@1 (%)                 | 89.2 | 89.9 | 90.8 | 90.8 | 91.0 |
> > > | Percent of Maximum Improvement   | 73.5 | 83.8 | 97.1 | 97.1 | 100.0 |
> > >
> > > > The authors also mention that they test on MATH 500 [...] please point me to where I can find the new experimental results
> > >
> > > The updated figures, including total FLOPs usage and downstream performance, are available [here](https://imgur.com/a/rhisaLK). Please refer to the third row in this set of images for the relevant results.

---

> > > > ### Author Response · Authors · 2024-11-27
> > > > **Following up!**
> > > >
> > > > Thank you for your review! Please let us know if further detail is needed or if the new experiments address your concerns.

---

### Author Response · Authors · 2024-11-23
**Overall Response**

We thank all the reviewers for their helpful feedback and suggestions. We appreciate that the reviewers recognize that our solution is simple and elegant (pws6, 6jZH), provides substantial cost reduction (pws6, 6jZH, Pqum), and that our paper is well-written (6jZH, Pqum).

In this rebuttal we make the improvements our methods offer more precise, concrete, and easy to understand. This was not clear to many reviewers as we made improvements to multiple aspects of Best-of-N which we measured in different ways.

Reviewers asked for evaluation with a more difficult benchmark that requires more reasoning, so we added evaluations with the MATH 500 benchmark. We also implemented a process reward model (PRM) baseline for requested comparison to our mid-generation self-evaluations. Finally, we added and restructured our figures with concrete metrics (FLOPs, wall-time) and additional ablations for improved clarity.

1. **What are we improving?**

Inference-time methods such as Best-of-N are expensive as they require multiple samples and an external reward model. The primary objective of our paper is to make this far more **cost-effective through the substantial reduction of total computation, memory usage, and energy consumption.** These metrics translate directly to cost in most modern LLM serving scenarios as queries are processed in continuous batches (Daniel et al, 2023) and minimal compute or memory is sitting idle. Our methods improve performance with fixed computation, memory, or energy.

We proposed methods that reduce cost without sacrificing latency (self-evaluations and early pruning) and methods that reduce more cost with some additional latency (adaptive sampling).

2. **How are we improving it?**

We introduce a generative reward model formulation, allowing LLMs to predict mid-generation the probability that restarting the generation will yield a better response. These capability aware and mid-generation self-evaluations enable adaptive inference-time compute strategies. Specifically, these predictions are obtained without an external reward model and can be used to decide whether or not to generate more samples, prune unpromising samples early on, or to pick the best sample.

We propose a two strategies to take advantage of this capability, namely adaptive sampling and early pruning. Adaptive sampling takes advantage of the capability-aware aspect of the self-evaluations to determine if generating more samples is beneficial. Early pruning takes advantage of mid-generation self-evaluations to stop unpromising samples early in generation.

Note that these new strategies we propose are primitives that could be combined or enhanced in various ways for potentially greater efficiency gains. To improve adaptive sampling specifically, we also proposed a novel annealing schedule as well as exponentially increasing batch sizes.

3. **How much are we improving it by?**

We measure efficiency with respect to the number of FLOPs during inference (following Hoffman et al 2022). We also provide exact wall-clock times in a controlled environment using SGLang (Zheng et al.) on a consistent hardware setup (8×A100 GPUs) in Figure 3B. Below, we will discuss the efficiency gains we get from each component we propose.

Firstly, capability-aware self-evaluations alone cut FLOPs by 2x compared to using external reward models. This is because the KV cache can be shared during the evaluation process with infilling of our self-evaluation prompt (16 additional tokens).

Additionally, we see roughly a **4x reduction in total FLOPs** to match the performance of Best-of-16 by additionally leveraging early pruning. This is **without any additional latency**. Adaptive sampling achieves **roughly a 6-8x reduction in total FLOPs** with the tradeoff of roughly 2x latency (as batches are processed sequentially) as shown in Table 3 in our appendix. Using early pruning or adaptive sampling is a choice that can be made depending on constraints an application has on latency and computational efficiency. We provide the total FLOPs usage and downstream performance in the updated Figure 2 and Figure 3. **[Results](https://imgur.com/a/pdQ878w)** with the PRM baseline show that mid-generation self-evaluations are still roughly 2x more efficient simply due to KV cache.

Also, our proposed methods allow for savings in memory usage. Memory usage is determined by the number of active model parameters, as well as the size of the KV cache which is largely a function of the total number of tokens generated. Since we both remove the reward model (2x less total parameters to store) and reduce the total number of tokens generated (2x with early pruning and 4x with adaptive sampling), **the memory savings are roughly proportional to the savings in the total number of FLOPs**.

Since we save total FLOPs and memory usage, **we also proportionally save energy consumption** which accounts for about 5-10% of the total cost of operation for modern GPUs.

---

### Meta-Review · Area_Chair_eAVL · 2024-12-29

**Metareview:**

In this paper, the authors propose a new strategy for efficient inference-time improvement of LLMs. Specifically, the authors exploit adaptive sampling (selecting the N in best-of-N as a function of the prompt) + early pruning (pruning unpromising mid-generations) upon the learned generative reward model. The authors conducted empirical comparison on GSM8k and MATH 500 to demonstrate the benefits in terms of FOLPS vs. Performance.

The paper is well-motivated and easy-to-follow. The major issues raised by the reviewers mainly lies in about the novelty and significance. Specifically, using additional reward model to guide the inference-time generation and pruning is not novel, e.g., [1, 2, 3]. Moreover, the empirical study is limited.  Without discussion and comparison to these methods on more tasks, it is difficult to justify the claimed benefits.

I suggest the authors to take the comments from the reviewers into account to emphasize the differences and benefits to the existing methods, especially the generative reward effect in accelerating inference in practice.


[1] Mudgal, Sidharth, Jong Lee, Harish Ganapathy, YaGuang Li, Tao Wang, Yanping Huang, Zhifeng Chen et al. "Controlled decoding from language models." arXiv preprint arXiv:2310.17022 (2023).

[2] Sun, Haotian, Yuchen Zhuang, Wei Wei, Chao Zhang, and Bo Dai. "BBox-Adapter: Lightweight Adapting for Black-Box Large Language Models." arXiv preprint arXiv:2402.08219 (2024).

[3] Chakraborty, Souradip, Soumya Suvra Ghosal, Ming Yin, Dinesh Manocha, Mengdi Wang, Amrit Singh Bedi, and Furong Huang. "Transfer Q Star: Principled Decoding for LLM Alignment." arXiv preprint arXiv:2405.20495 (2024).

**Additional Comments On Reviewer Discussion:**

The authors provides detailed answers and additional experiments, which partially addressed the questions from the reviewers. However, there are still concerns remained from reviewer.

---

### Decision · Program_Chairs · 2025-01-22

Reject